# Variation in Nitrogen Utilization and Nutrient Composition across Various Organs under Different Strip Logging Management Models in Moso Bamboo (*Phyllostachys edulis*) Forest

**DOI:** 10.3390/plants13111448

**Published:** 2024-05-23

**Authors:** Bingjun Li, Linzheng Xu, Wenchen Chen, Yanmei Pan, Tianyou He, Liguang Chen, Jundong Rong, Yushan Zheng

**Affiliations:** 1College of Forestry, Fujian Agriculture and Forestry University, Fuzhou 350002, China; fafulbj@163.com (B.L.); 13023866670@163.com (L.X.); cerdwin2003@163.com (W.C.); panyanmei@163.com (Y.P.); clguang_cn@163.com (L.C.); zys1960@163.com (Y.Z.); 2College of Landscape Architecture and Art, Fujian Agriculture and Forestry University, Fuzhou 350002, China; hetianyou1985@163.com

**Keywords:** moso bamboo, strip logging, ^15^N isotope tracing, nitrogen utilization, nutrient content

## Abstract

The rapid restoration and renewal of the moso bamboo logging zone after strip logging has emerged as a key research area, particularly regarding whether nutrient accumulation and utilization in reserve zones can aid in the restoration and regeneration of the logging zone. In this study, a dynamic ^15^N isotope tracking experiment was conducted by injecting labeled urea fertilizer into bamboo culms. Logging zones and reserve zones of 6 m, 8 m, and 10 m widths were established. The conventional selective logging treatment served as a control (Con). Measurements were taken in May and October to assess the differences in nitrogen accumulation ability, utilization rates, and nutrient content across different organs in bamboo forests at different growth stages and under different treatments. Principal component analysis was conducted to evaluate and determine the importance of each indicator and strip logging treatment comprehensively. The results showed that various bamboo organs exhibited higher nitrogen accumulation and utilization rates during the peak growth period compared to the late growth period. Leaves had the highest nitrogen accumulation and utilization rates than the other organs. The average C content in various bamboo organs under different logging treatments exhibited subtle differences, irrespective of variation in logging width treatments. Bamboo culm exhibited the highest carbon accumulation. The C content in various bamboo organs was higher during the peak growth period than in the late growth period. The nitrogen content peaked in the leaves during the two growth stages and was significantly higher compared to the other organs. Most bamboo organs in the logging zones exhibited relatively higher nitrogen content than in the reserve zone and Con group. The P content was highest in bamboo leaves compared with other organs across the different strip logging treatments. Principal component analysis revealed relatively high absolute values of the coefficients for the C content, bamboo stump C content, and culm Ndff%. Log8 and Res10 zones had the highest comprehensive evaluation scores, indicating that Log8 and Res10 had the best effect on the promotion of nitrogen utilization and nutrient accumulation in various organs of moso bamboo.

## 1. Introduction

*Phyllostachys edulis* holds significant economic and ecological importance in China [1]. However, an increase in logging costs due to labor shortage poses a practical challenge to the cultivation and industrial development of bamboo forests [2]. Consequently, a strip logging method tailored for bamboo forests has been proposed to reduce logging costs and achieve sustainable management. Studies report that strip logging is an important silvicultural and disturbance mechanism, directly affecting stand productivity [3,4]. Well-managed logging activities can optimize forest stands, enhance microenvironment conditions, conserve water sources, prevent soil erosion, and promote ecological balance while generating valuable forest by-products [5]. Poor logging activities reduce forest productivity [6,7]. Wang demonstrated that the shoot production per unit area in logging zones was higher than that of the control plot, but the average diameter at breast height and biomass per unit area of newly grown bamboo were lower compared to the control plot [8]. Zhang observed that logging promoted species richness, especially in the 8 m and 15 m logging zones [9]. Notably, strip logging significantly enhanced understory vegetation biomass in the 8 m and 15 m logging zones. In addition, studies have elucidated the relationships between nutrients in bamboo stands and the growth of associated vegetation and strip logging [10,11,12]. Strip logging significantly affects soil nutrients and nutrients in various tree components [13]. Appropriate logging practices are essential for maintaining a balanced nutrient supply in bamboo stands and promoting the efficient growth of bamboo forests.

Nitrogen is one of the important limiting nutrients in terrestrial ecosystems, and its soil nutrient availability directly affects the structural and functional stability of ecosystems [14,15], while differences in N content and utilization in plants can significantly affect plant growth and functional characteristics [16]. Gordon showed that the fine root N content was negatively correlated with the diameter of fine roots, with smaller diameter fine roots having higher N content and their function being dominated by nutrient and water absorption [17]. The N content of leaves also determines the differences in photosynthetic capacity of plant leaves, and a number of studies have shown a positive correlation between the N content of leaves and photosynthetic rate, respiration rate, and productivity [18,19,20]. Leaves of fast-growing, short life-history herbaceous plants contain higher levels of N, while leaves of slow-growing, long life-history evergreen plants contain lower levels of N [21,22,23]. Nitrogen uptake by plants mainly comes from fertilizer nitrogen and soil nitrogen, and with ^15^N as a stable isotope, the ^15^N isotope tracer method can distinguish between fertilizer nitrogen and soil nitrogen uptake by plants and accurately traCon and detect the transformation and direction of fertilizer N after it is applied into the soil, which can truly reflect the state of the actual use of fertilizers by plants [24,25,26].

Bamboo is a typical clonal plant with sustained nitrogen utilization capacity. The ^15^N isotope tracing technology can be used to explore the accumulation and utilization of nutrients by bamboo and provide an accurate and comprehensive understanding of nitrogen accumulation, utilization, and distribution across different plant parts and growth stages. This method minimizes interference from other nitrogen sources, providing a robust foundation for studying plant nitrogen utilization and accumulation. Therefore, whether different logging width treatments will have significant effects on nitrogen accumulation, utilization and elemental content in various organs of moso bamboo has become a more popular topic of discussion. Some scholars have proposed that wider harvesting strips will increase the number of understorey vegetation species, leading to increased competition for nutrients and thus reducing the accumulation of nitrogen in bamboos [27]; others believe that harvesting strips that are too small will result in insufficient retention strip spacing and have no significant effect on nitrogen accumulation and nutrient content in bamboos [28]. It was also suggested that too small a logging zone would result in insufficiently spaced retention zones and would have no significant effect on bamboo nitrogen accumulation and nutrient content.

In this study, experiments were conducted in a pure bamboo forest in Zhangping City, Fujian Province. Three strip logging treatments with widths of 6 m, 8 m, and 10 m were established. The ^15^N tracing technology and elemental determination methods were integrated to assess the accumulation capacity and utilization efficiency of fertilizer nitrogen in various bamboo organs under different logging zones with varying logging width treatments during two growth periods. To elucidate the differences in nitrogen accumulation, utilization capacity, and elemental content of various organs of moso bamboo in the strip logging and reserve zones under different strip-harvesting treatments and the laws of dynamic changes.

## 2. Materials and Methods

### 2.1. Overview of the Experimental Site

The study area is located in Zhangping, Longyan City, Fujian Province (117°11′–117°44′ E and 24°54′–25°47′ N), popularly known as the “Chinese characteristic bamboo town”. The area is located in the upper reaches of the Jiulong River in the southwest of Fujian Province, characterized by low mountains and hills and altitudes ranging from 610 to 880 m. The region experiences a subtropical marine monsoon climate characterized by warmth, humidity, and sufficient rainfall. The area has warm weather during winter and cool temperatures during summer, with a significant vertical climate and distinct dry and wet seasons. The experimental site specifically lies within Zhangping City, with an average annual temperature of 18.8 °C, an average annual precipitation ranging from 1450 to 2100 mm, an average annual sunshine duration of 1853 h, and a frost-free period between 251 and 317 days. The main vegetation types under the bamboo canopies are *Cunninghamia lanceolate* and dense herbaceous species such as *Boehmeria nivea* and *Dicranopteris dichotoma*. The region has yellow-red soil, with a slope of 25°–30° and a soil layer depth spanning 60 to 80 cm. The forest landscape primarily comprises pure bamboo forest, managed through fertilization, weed control, and removing unhealthy vegetation.

### 2.2. Methods

#### 2.2.1. Plot Establishment and Experimental Design

In mid-February 2020, we investigated the stand factors and vegetation conditions of the sample plots in the study area and measured the diameter at breast height (DBH), tree height, height under branches, and density of all moso bamboo forests in the study area by the method of cheConing the ruler of each tree and calculating the mean values, which resulted in the average density of moso bamboo forests ranging from 2756–3438 hm^−2^, with an average DBH of 8.66 cm, and the height under branches and plant heights of 4.89 m and 11.37 m, respectively. The average diameter at breast height was 8.66 cm, and the height under branch and plant height were 4.89 m and 11.37 m, respectively. Experiments on the strip logging management methods were conducted within pure bamboo stands with consistent management practices, similar slopes, and a generally uniform stand structure. As the selected sample site is a pure forest of moso bamboo, there are no other species of trees in the sample site that were harvested by the traditional selective logging method in the early stage. Strip logging treatments started in March 2020, incorporating three types of strip logging treatments distinguished by logging widths: Log6 (6 m), Log8 (8 m), and Log10 (10 m). The vertical logging length was 15 m, resulting in three standard plots with distinct logging widths and areas. Three blocks were designated within each type of standard plot, and a reserve zone (no logging area) was established between adjacent standard plots. These reserve zones, labeled Res6, Res8, and Res10, had the same width and area as the adjacent logging zone. A control plot (Con) was established 200 m from the logging treatment area to minimize the influence of edge effects from strip logging. The area specification of the Con was 20 m × 20 m, with a total of 3 control plots established.

Fertilization was carried out in March 2021, and standard bamboos were selected in different bandwidths of the harvesting zone, the retention zone and the control group according to the upper, middle and lower slope positions, as far as possible located in the middle of the logging widths (Figure 1 and Figure 2), in order to minimize the edge effect of other logging width treatments affecting the accuracy of the experimental results, and one standard bamboo was selected for the bamboo cavity injection at each slope position, and three bamboo plants were selected at each logging width treatment site, and the bamboos were marked by spraying paint on the surface of bamboos in order to differentiate them. The bamboo surface was marked with spray paint for differentiation. The bamboo selection and labeling method of the control group was the same as that of the reserve zones and strip logging zones, and a total of 63 bamboos were injected and labeled. This method involved drilling a downward hole into the bamboo cavity with a 0.5 cm diameter drill. The hole is drilled at the base of each plant, approximately 10 cm from the ground level. Subsequently, a medical syringe was used to inject 20 mL of 0.45% urea labeled with ^15^N into each bamboo plant, and the hole was sealed with sealing mud. The nitrogen content of the ^15^N labeled urea was approximately 46.67%, with ^15^N isotopic abundance at 30%. The urea was purchased from the Shanghai Research Institute of Chemical Industry.

#### 2.2.2. Sample Collection and Analysis

Samples were collected during two distinct periods: May 2021 (peak growth period) and October 2021 (late growth period). Within each of the 6 m, 8 m, and 10 m logging zones and reserve zones, three standard bamboo plants were selected from the upper, middle, and lower slope positions, with 18 standard plants. These plants were carefully cut down at ground level, and their stumps were excavated. Subsequently, leaves, branches, culms, stumps, and rhizomes were separated. Tissue samples weighing between 200 and 500 g for each organ were collected and placed into sample bags in an icebox for analysis in the laboratory.

Samples were transported to the laboratory and subjected to desiccation in an oven at 105 °C for 30 min. Subsequently, the samples were dried at a temperature of 60 °C to obtain a constant weight. The dried samples were pulverized into powder using a small milling machine. The powdered samples were sieved through a 100-mesh sieve. A portion of the sample powder was reserved for ^15^N determination, whereas the other portion was used to determine C, N, and P contents. Strict measures were implemented during the sample preparation process to prevent contamination of unlabeled samples with ^15^N-labeled samples. The abundance of ^15^N in plants was measured using a stable isotope ratio mass spectrometer (Isoprame 100, UK). Leaves, branches, culms, stumps, and rhizomes were separately washed, dried, and pulverized. Total carbon content was determined using the potassium dichromate external heating method [29], whereas total nitrogen content was determined using the Kjeldahl nitrogen determination method [30]. The total phosphorus content was assessed using the alkali-melt molybdenum-antimony anti-colorimetric method [31].

The ^15^N content in each organ was determined using the equations below:

The proportion of ^15^N absorbed by the sample to the total nitrogen in the sample organs (Ndff%) was expressed as Ndff% = (^15^N atom % excess in the sample)/(^15^N atom % excess in the fertilizer) × 100%
Fertilizer utilization efficiency (^15^NUE) = ^15^N absorption amount (g)/^15^N application amount (g) × 100%
Atom percentage excess = sample abundance − natural abundance
where Ndff% represents the percentage of nitrogen derived from fertilizers relative to the total nitrogen content of plants, which is referred to as the nitrogen accumulation capacity [32]; ^15^NUE (^15^Nuse efficiency) denotes the percentage of fertilizer N utilization [33].

### 2.3. Data Processing and Analysis

Excel 2016 and the SPSS 22.0 statistical package were used for data processing and analysis. One-way ANOVA and Duncan analysis were performed for multiple comparisons, with letters used to indicate significance. The significance level was set at α = 0.05. Principal component analysis (PCA) was conducted to identify the dominant factors, and a comprehensive ranking of logging treatments with different logging widths was performed. Plots were generated using Origin2022 software.

PCA is a multivariate statistical analysis method that involves a linear transformation of multiple variables to select a reduced set of important variables. The main steps of performing PCA include: (1) standardization of the original data by subtracting the mean and dividing by the standard deviation; (2) computation of the covariance matrix between multiple variables using the normalized data; (3) identification of eigenvalues and eigenvectors, which decompose the covariance matrix to obtain eigenvalues; (4) principal component construction, which involves the determination of the number of principal components based on eigenvalues and eigenvectors; and (5) calculation of the comprehensive scores of each indicator.

## 3. Results

### 3.1. Various Strip Logging Treatments Exhibit Different Effects on the Nitrogen Accumulation Ability of Various Organs in Bamboo

As shown in Figure 3, the nitrogen accumulation capacity of moso bamboo leaves, branches, culms, and stumps in the reserve zone was significantly higher in different strip logging treatments at the peak growth stage (May) than at the end growth stage (October). In the logging zone, the differences were larger, and the nitrogen accumulation capacity in all organs of moso bamboo under Log6 treatment peaked in October. The nitrogen accumulation capacity of bamboo culms and stumps in Log8 and Log10 treatments was highest in May, while leaves and stumps peaked in October. Differences in nitrogen accumulation capacity of bamboo rhizomes among strip logging treatments were small and did not reach the level of significance in May and October. This indicates that the nitrogen accumulation capacity in most organs of moso bamboo increased significantly in the reserve zone during the peak growth stage, while the nitrogen accumulation capacity of different organs in the logging zone varied greatly between the two periods.

In May, the nitrogen accumulation capacity of moso bamboo leaves, branches, culms, and stumps in the reserve zone of different logging widths was significantly higher than that of the logging zone and the Con under the same logging width, with the highest nitrogen accumulation capacity in the leaves in the reserve zone, which had an average Ndff% of 8.28%. The nitrogen accumulation capacity of bamboo rhizomes in the reserve zone was slightly higher than that in the logging zone, but the overall difference was not significant. In contrast, the nitrogen accumulation capacity of organs in the reserve zone was overall significantly lower than that of the logging zone and the Con in October, with the highest nitrogen accumulation capacity in the leaves of the logging zone, which had a mean Ndff% of 4.08%. This indicates that the strip logging treatment promoted the nitrogen accumulation capacity of various organs of moso bamboo in the reserve zone at the peak growth stage and the logging zone at the end of the growth stage.

### 3.2. Different Strip Logging Treatments Have Varying Effects on Nitrogen Utilization Efficiency in Various Bamboo Organs

Different strip logging treatments had significantly different impacts on the nitrogen utilization efficiency (^15^NUE%) in various bamboo organs at different stages (Figure 4). In May, all organs’ nitrogen utilization of moso bamboo in the logging zone was lower than in October, with the most significant difference in nitrogen utilization in leaves. Whereas, the overall nitrogen utilization of all organs in the reserve zone peaked in May, with the highest mean nitrogen utilization of 9.57% in leaves. Nitrogen utilization of leaves, branches, culms, and bamboo rhizomes in Con was highest in May, while the opposite was true for bamboo stumps, where nitrogen utilization peaked in October. This indicates that all organs’ nitrogen utilization of moso bamboo in the reserve zone was significantly higher at the peak growth stage than at the end of the growth stage, while the opposite trend was observed in the logging zone.

In May, all organs’ nitrogen utilization of moso bamboo in different bandwidth retention strips was higher than that in the same logging width treatments, with the highest nitrogen utilization in the leaves in the reserve strips. The mean values of all organs’ nitrogen utilization in Res8 and Res10 were higher than those in Res6. In contrast to Con, nitrogen utilization in branches, culms, and bamboo rhizomes was higher in Con than in the logging and reserve zones, while the opposite trend was observed in stumps. In October, the nitrogen utilization of various organs showed the opposite trend to that of May, with the mean nitrogen utilization of various organs in the logging zone being higher than that in the reserve zone and Con, and the overall difference in nitrogen utilization of leaves being the most significant. This indicates that the nitrogen utilization of various organs in the harvested and preserved zones of moso bamboo showed opposite trends in different growth periods.

### 3.3. Different Strip Logging Treatments Exhibit Varying Effects on the Contents of Key Elements in Various Bamboo Organs

#### 3.3.1. Different Strip Logging Treatments Affect the Contents of Key Elements in Bamboo Leaves

As shown in Figure 5, the carbon (C) and nitrogen (N) contents of moso bamboo leaves in the different strip logging treatments showed relatively similar trends over the two growth periods, with both the C and N contents of leaves in the reserve zone peaking in May and the C and N contents of leaves in the logging zone reaching a maximum in October, with the C contents of leaves in Res8 and Log8 peaking in both May and October, respectively. The mean N content in the logging zone was higher than that in the reserve zone in both periods, reaching 27.28 g·kg^−1^ in May and 31.61 g·kg^−1^ in October, which was 17.7% and 39.4% higher than that in the reserve zone in the same period. The C contents of leaves in Con were overall lower than the different strip logging treatments in both periods, while the N content was at the average level in both periods, and overall differences were not significant.

The phosphorus (P) contents of moso bamboo leaves in the reserve zone, logging zone, and Con treatment all peaked in May and were all significantly higher than in October. In contrast, the P contents in the logging zone were higher than those in the reserve zone and Con, with mean values reaching 3.08 g·kg^−1^ and 1.41 g·kg^−1^ in May and October, respectively, which were 49.0% and 11.9% higher than those in the reserve zone, and 44.8% and 2.9% higher than those in the Con at the same period. It can be seen that the P contents of moso bamboo increased significantly in the logging zone during the growth period, while the overall difference in P contents at the end of the growth period was small.

As shown in Table 1, in both periods, the C/N of moso bamboo leaves in the reserve zone was significantly higher than that in the logging zone and Con. In October, C/N in the logging zone of different widths was lower than Con, indicating that the reserve zone had a boosting effect on the leaf C/N of moso bamboo at different growth periods, and a boosting effect on the leaf C/N of moso bamboo in the logging zone at the peak growth period, but an inhibitory effect at the end of the growth period.

In May, N/P of moso bamboo leaves was higher in the Con than in the strip logging treatments, and the highest in the strip logging treatments was Res8 at 13.13, which was 57.8% higher than that of Log8 in the same logging width. In October, the N/P of moso bamboo leaves was higher than that in May in different strip logging and Con treatments, and the N/P was higher in the logging zone than in the reserve zone, with the highest in Log10, which was 20.8% higher than that in Con.

#### 3.3.2. Different Strip Logging Treatments Have Varying Effects on the Elemental Composition of Bamboo Branches

As shown in Figure 6, in May, the C contents of moso bamboo branches in the logging zone were lower than those in the Con and reserve zones of the same width, and there was no significant difference in the C contents of branches under all treatments of strip logging in October. In May, the N contents of moso bamboo branches under the strip logging treatments were lower than those of the Con, and the N contents of the logging zone were higher than those of the reserve zone, with Log6 peaking at 6.66 g·kg^−1^, which was 41.1% higher than that of Res6. In October, branch N contents under all treatments of strip logging were higher than that of Con, with the highest branch N content of 8.07 g·kg^−1^ in Log8, which was 19.9% and 33.4% higher than that of Res8 and Con, respectively. Branch N contents were higher in October than in May in all treatments of strip logging, and the opposite was true for Con, which peaked in May.

In the May strip logging treatments, the P content of branches was lower than Con in all treatments except Res6 and Log8, with the highest P content of 2.05 g·kg^−1^ in Log8 branches, which was 31.4% higher than Con. In October, the P content of moso bamboo branch was lower than that of Con under both strip logging treatments, and the P contents of branch in the logging zone were higher than that of the reserve zone, but the differences in the P contents of branch under the different strip logging treatments and Con treatments did not reach the level of significance in all cases.

As shown in Table 2, in May, the C/N of moso bamboo branches under strip logging treatments were all higher than that of the Con, and the C/N of branches in the reserve zone were all higher than that of the same width logging zone, with the highest branch C/N of 115.68 under the Res8, which was 75.6% higher than that of the Con. In October, the C/N of moso bamboo branches showed a similar trend to that of May, with all the reserve zone higher than the treatments with the same width of the logging zone, but the branch C/N of all treatments except Res6 was smaller than that of the Con, suggesting that strip logging treatments have a certain inhibitory effect on the C/N of the branches at the end of the growth period.

The N/P of moso bamboo branches under the strip logging treatments were all lower than that of the Con in May. The highest N/P of Res10 branches in the strip logging treatments was 5.36, which was 26.1% higher than that of Log10 and 23.3% lower than that of the Con, and the overall N/P of moso bamboo branches in the reserve zone was higher than that of the logging zone during this period. In October, on the other hand, the N/P of moso bamboo branches under strip logging treatments showed an opposite trend to that of May, both being higher than that of the Con, while the N/P of branches in the logging zones were all slightly higher than that of the same width of the reserve zones, with Log10 being the highest at 8.70, which was 2.2% and 40.3% higher than that of the same period in Res10 and Con, respectively.

#### 3.3.3. Different Strip Logging Treatments Have Varying Effects on the Element Content in Bamboo Culms

As shown in Figure 7, the C contents of both Log6 and Log10 culms in the logging zone were lower than those of the same width of the reserve zone in May, with the highest culm C contents of 630.00 g·kg^−1^ in the Res10, which was 11.2% higher than that of the same period in Con. In October, the contents of Culm in Con were higher than in the strip logging treatments and higher in the reserve zone than in the logging zone of the same width, with Res8 being the highest at 572.70 g·kg^−1^. Overall, the C contents of culm under the strip logging treatments were generally not significantly different between the two periods.

In May, the N contents of moso bamboo culm in the logging zones were lower than those in the same width of the reserve zones, but most of the strip logging treatments had fewer N contents in the bamboo culms than the Con, with the Log10 reaching the highest level of 4.85 g·kg^−1^, which was slightly higher than that of the Con in the same period by 1.5%. The N contents of moso bamboo culms were higher than those of the Con in the strip logging treatments in October, with Res10 being the highest, 58.9% higher than that of the Con in the same period. In short, there was little overall difference in the N contents of moso bamboo culms in the strip logging treatments in the two periods.

In May, the overall mean P contents of culms were higher in the logging zones than in the reserve zones. The P contents of moso bamboo culms were higher in all logging zones than in the same width reserve zones in October, with Log8 being the highest at 0.97 g·kg^−1^, which was 12.8% higher than that of the Con, but the P contents of moso bamboo culms under both strip logging and Con treatments in October were less than those in May.

As shown in Table 3, in May, the C/N of culms in the reserve zones was higher than that of the same width logging zones and the Con, with the Res8 being the highest at 187.70, which was 58.4% higher than Con. In October, the C/N of culms was lower than Con in all strip logging treatments except Res6, and the C/N of culms differences were not significant in most treatments in October and May. From the differences in N/P characteristics of moso bamboo culms at different periods of time, in May, N/P of moso bamboo culms was lower than Con under all strip logging treatments except Log10, with culms in Res10 having the highest N/P of 5.50. In October, it was significantly higher than Con by 51.1%, and the C/N of culms in both strip logging and Con were higher in October than in May, with some reaching a significant level.

#### 3.3.4. Different Strip Logging Treatments Have Varying Effects on the Elemental Composition of Bamboo Stumps

As shown in Figure 8, in May, the C content of bamboo stumps was generally higher in the reserve zone than in the logging zone, but most of the differences were not significant, with the highest bamboo stump C content of 592.61 g·kg^−1^ in Res10, which was 7.8% higher than that in Con. In October, the C content of bamboo stumps was higher than Con under all strip logging treatments except Res10, with Res6 being the highest. The C content of bamboo stumps in both the strip logging and Con treatments was lower in October than in May, but most did not reach significance.

In May, the N content of stumps was significantly higher under all strip logging treatments than Con and was generally higher in the reserve zone than in the logging zone, except for Res8. The N content of stumps was significantly higher in October under all strip logging treatments except Res6, and the N content was significantly higher in October than in May.

The P content of stumps was significantly higher than that of Con under all strip logging treatments in May, while the logging zones were generally higher than the reserve zones, with the highest P content of 2.05 g·kg^−1^ in Log8, which was 113.5% higher than that of Con. In October, the P content of stumps under all strip logging and Con treatments was generally not significantly different from that in May, with the highest being Log8 at 1.79 g·kg^−1^, which was 64.2% higher than the Con.

As shown in Table 4, the C/N of bamboo stumps under each strip logging treatment was generally lower than Con in both periods, except for Res6. The C/N of bamboo stumps under all strip logging and Con treatments was significantly lower in October than in May. The N/P of bamboo stumps in the reserve zone was higher than that of the same-width logging zone and the Con treatment, while the N/P of stumps in the logging zone was lower than that of the Con treatment, with the highest being 7.16 in the Res10. The N/P of bamboo bamboos was higher in October than that in May in all treatments.

#### 3.3.5. Different Strip Logging Treatments Exhibit Varying Effects on the Elemental Composition of Bamboo Rhizomes

As shown in Figure 9, the C content of rhizomes in the reserve zone was generally higher than that in the logging zone and Con in May, with the highest C content of 617.75 g·kg^−1^ in the Res6. In October, the C content of rhizomes was significantly lower under all strip logging treatments than in May, with the highest C content in the reserve zones, but none of the differences in the C content of rhizomes between all treatments reached a significant level.

The N content of rhizomes was lower than that of the Con in May in all strip logging treatments, and the overall N content was higher in the reserve zone than in the logging zone. In October, the N content of rhizomes under most of the treatments was higher than that in May and higher than that of Con in the same period, with the highest being 11.24 g·kg^−1^ in Log8, which was significantly higher by 70.3% than that of Con.

In May, the difference in the P content of rhizomes between most of the strip logging treatments and Con was not significant, and the P content of rhizomes was lower under all strip logging treatments than Con except for Log8. The P content of rhizomes under each treatment showed a similar trend in both periods.

As shown in Table 5, the C/N of rhizomes under the logging zone was generally higher than that of the reserve zone and significantly higher than that of the Con in May, with Log6 being the highest at 120.17. In October, the C/N of rhizomes was significantly lower under all strip logging treatments than in May, and all treatments were lower than Con except Res6 and Res8. The C/N of rhizomes in all strip logging treatments were lower than Con in May, with the highest rhizomes N/P of 6.54 in Res8 among the strip logging treatments. In October, the C/N of rhizomes was higher than Con in all strip logging treatments, partially reaching significant differences.

### 3.4. PCA of Nitrogen Utilization and Nutrient Content in Various Organs of Moso Bamboo under Different Strip Logging Treatments

PCA was conducted on the nitrogen (N) utilization and nutrient content of 25 bamboo organs (Table 6), and the top 5 principal components were selected. The top five principal components (PC1, PC2, PC3, PC4, and PC5) explained 38.557%, 32.678%, 12.618%, 8.264%, and 5.242% of the variance in the nitrogen (N) utilization and nutrient content in the 25 bamboo organs. The cumulative variance contribution rate of the top five principal components was 97.358% of the variance, indicating that they explained all the variance in the data. In the first principal component, relatively large absolute coefficients of 0.954, 0.950, and 0.919 were estimated for the C content in bamboo culms, stumps, and culms Ndff%, respectively. This indicates that these indicators are important in reflecting the N utilization and nutrient content in various organs of moso bamboo under different strip logging treatments.

Five principal component expressions can be derived based on the eigenvalues and eigenvectors of the five principal components as follows:Y_1_ = 0.781X_1_ + 0.702X_2_ + 0.919X_3_ + 0.744X_4_ + 0.813X_5_ + 0.622X_6_ + 0.415X_7_ + 0.321X_8_ + 0.537X_9_ + 0.805X_10_ + 0.804X_11_ − 0.561X_12_ − 0.155X_13_ + 0.865X_14_ − 0.132X_15_ + 0.182X_16_ + 0.954X_17_ + 0.036X_18_ + 0.145X_19_ + 0.950X_20_ + 0.251X_21_ + 0.049X_22_ + 0.676X_23_ + 0.518X_24_ + 0.685X_25_
Y_2_ = −0.483X_1_ − 0.559X_2_ − 0.037X_3_ − 0.206X_4_ − 0.236X_5_ − 0.504X_6_ − 0.711X_7_ + 0.685X_8_ + 0.633X_9_ + 0.516X_10_ − 0.322X_11_ + 0.813X_12_ + 0.900X_13_ − 0.079X_14_ + 0.859X_15_ + 0.464X_16_ − 0.119X_17_ + 0.899X_18_ + 0.595X_19_ − 0.029X_20_ + 0.667X_21_ + 0.727X_22_ − 0.471X_23_ + 0.761X_24_ + 0.379X_25_
Y_3_ = −0.135X_1_ + 0.290X_2_ − 0.292X_3_ + 0.578X_4_ + 0.262X_5_ − 0.174X_6_ + 0.372X_7_ − 0.642X_8_ + 0.182X_9_ + 0.142X_10_ − 0.123X_11_ − 0.051X_12_ + 0.284X_13_ − 0.107X_14_ − 0.084X_15_ + 0.648X_16_ − 0.236X_17_ − 0.365X_18_ + 0.663X_19_ − 0.231X_20_ − 0.213X_21_ + 0.566X_22_ + 0.425X_23_ − 0.268X_24_ − 0.306X_25_
Y_4_ = −0.266X_1_ − 0.262X_2_ − 0.005X_3_ + 0.145X_4_ + 0.224X_5_ − 0.496X_6_ − 0.297X_7_ + 0.072X_8_ + 0.513X_9_ − 0.035X_10_ + 0.417X_11_ − 0.075X_12_ + 0.104X_13_ − 0.056X_14_ − 0.209X_15_ − 0.494X_16_ + 0.006X_17_ + 0.134X_18_ + 0.108X_19_ + 0.184X_20_ + 0.639X_21_ + 0.226X_22_ + 0.077X_23_ − 0.188X_24_ − 0.452X_25_
Y_5_ = −0.030X_1_ + 0.190X_2_ + 0.263X_3_ + 0.017X_4_ − 0.183X_5_ − 0.018X_6_ + 0.301X_7_ + 0.097X_8_ − 0.041X_9_ − 0.105X_10_ + 0.146X_11_ + 0.087X_12_ − 0.262X_13_ − 0.473X_14_ + 0.433X_15_ − 0.237X_16_ + 0.107X_17_ + 0.194X_18_ + 0.418X_19_ + 0.090X_20_ − 0.148X_21_ − 0.283X_22_ + 0.273X_23_ − 0.123X_24_ − 0.262X_25_

The normalized raw data was substituted into the principal component expressions to calculate the comprehensive principal component scores for each strip logging treatment (Table 7). Treatments Log8 and Res10 exhibited higher comprehensive evaluation scores, indicating that the logging zone Log8 and the reserve zone Res10 were more favorable for N utilization and nutrient accumulation in various organs of moso bamboo.

## 4. Discussion

The alteration in the habitat of moso bamboo due to strip logging affects the accumulation capacity of fertilizer nitrogen in various organs of moso bamboo, reflecting changes in the ecological environment. Moso bamboo, characterized by clonal integration capabilities, can alleviate the impact of strip logging and thus exhibit rapid recovery after logging. Investigating the accumulation ability of fertilizer nitrogen in various bamboo organs under different logging treatments and exploring the nutrient transport and accumulation capacities of bamboo plants under logging zones with different logging widths are crucial for identifying fertilization targets and determining rational amounts of nutrients. The nitrogen absorbed by bamboo primarily originates from soil, artificial fertilization, and atmospheric deposition in its natural environment. Several studies have demonstrated the efficacy of ^15^N tracing technology in determining the amount of nitrogen absorbed by plants from these three sources [34,35]. In this study, the nitrogen accumulation capacity of moso bamboo organs in the reserve zone increased significantly during the growth stage, while the nitrogen accumulation capacity of different organs in the logging zone varied considerably between the two periods, and the nitrogen accumulation capacity of organs was generally higher than that of 10 m when the logging widths were 6 m and 8 m, respectively. This may be due to the fact that when the logging width is large, the bamboo system needs to consume a large amount of energy to satisfy the rapid growth of bamboo rhizomes when using them for nutrient transport over long distances, and a large amount of N is involved in cell division and growth, the synthesis of proteins and carbohydrates, etc. In order to adapt to changes in the external environment, excessive nitrogen consumption also leads to a decrease in the capacity of organ nitrogen accumulation, which is similar to Zheng et al. [36]. In addition, compared with Con, the nitrogen accumulation capacity of moso bamboo organs in the reserve zone at the peak growth stage and in the logging zone at the end of the growth stage was significantly promoted. This may be due to the fact that in the reserve zone, because the number and species of understory vegetation did not change, its growth cycle pattern was not affected too much, so the nitrogen accumulation capacity of each organ of moso bamboo would be significantly higher in the growth stage to accumulate more nitrogen to meet the growth demand. In contrast, the logging zone in the strip-logging treatments enhanced the light duration and quality of the forest floor due to the reduction of the number and species of understorey vegetation, and the competitive pressure for growth resources was reduced, which favored the enhancement of photosynthesis of the new bamboo on the forest floor. This may have led to the postponement of the peak of the nitrogen accumulation capacity of the moso bamboo, so that the nitrogen accumulation capacity of the moso bamboo on the harvesting zone reached its peak at the end of the growth period. The results of the study were consistent with those of Zhang Yangyang et al. [37] and Elser et al. [38].

Nitrogen utilization of moso bamboo organs was significantly higher in the reserve zone at the peak growth stage than at the end of growth, while the opposite trend was observed in the logging zone, with the peak of nitrogen utilization of all organs occurring at the end of growth. This also suggests that the strip logging treatments can change the occurrence time of peak nitrogen utilization in the growth cycle of moso bamboo because the strip logging changed the growth environment of the original moso bamboo pure forest, the logging zone made the resource pressure in the early stage reduced due to the reduction of weed and tree species and number, and the growth bloom did not need excessive N conversion efficiency, so its nitrogen utilization was lower than that of the reserve zone and the Con in this stage, which is similar to previous studies [39]. In addition, this study revealed that the leaves of moso bamboo exhibited a higher nitrogen utilization efficiency under different logging treatments compared with other aboveground organs, with the efficiency gradually increasing over time. Bamboo stands typically demonstrate alternate bearing characteristics, producing a heavy crop in one year (on-crop), followed by a lighter crop (off-crop) the next year. During the on-crop period, bamboo stands primarily undergo sprouting and bamboo formation stages, then transition to the off-crop period. Logging during the on-crop period results in an increased utilization rate of fertilizer nitrogen for the leaves of young bamboo growing during the off-crop years. Bamboo rhizomes exhibit slow growth in the early stages and rapid growth in the later stages after the formation of the new bamboo. Bamboo stumps serve as underground nutrient absorption and storage systems, exhibiting a significant demand for N fertilizer [40]. After transitioning into the peak growth period, the bamboo stands under the Log8 treatment released a significant amount of bamboo leaves as the rhizomes grew, gradually becoming the primary organ for photosynthesis. This results in increased leaf chlorophyll content and improved photosynthetic ability, leading to a higher demand for nitrogen. Consequently, bamboo leaves demonstrated the highest fertilizer N utilization efficiency across all the logging treatments [41]. In heterogeneous environments, clonal ramets distributed in resource-rich patches often act as donors, whereas clonal ramets located in resource-poor patches act as receptors. Resource heterogeneity is the main factor that promotes integration. Under the strip logging management model, habitat heterogeneity levels varied depending on the different logging widths of logging zones, leading to differences in the nitrogen utilization ability of various bamboo organs across different logging treatments. Notably, under the spatial heterogeneity framework induced by the Log8 logging width treatment, the nitrogen utilization abilities of various bamboo organs were similar to those under the control treatment.

Mineral elements constitute the building bloCons of living organisms, with C, N, and P being vital for plant growth and serving as the primary components of plants. In this study, no significant difference in the average C content of various bamboo organs was observed under the different logging treatments with varying logging widths. Bamboo culms exhibited the highest carbon accumulation, indicating that bamboo allocates more carbon in the culm during strip logging treatments. This phenomenon can be explained by the hypothesis that moso bamboo preferentially allocates more C in culms to promote culm growth and lignification in response to heterogeneous spatial environments induced by logging treatments. Consequently, this improves height growth and lodging resistance and facilitates a more significant acquisition of light resources to enhance photosynthesis intensity [42]. The study revealed that under a logging width of 6 m in the Log6 and Res6 zones, the overall carbon accumulation in various organs was lower compared to other logging width treatments and Con. Conversely, under a logging width of 8 m, the highest C contents were observed in Log8 and Res8 treatments. The C content in various bamboo organs was higher during the T1 period than that observed during the T2 period. The C content in various organs of branches and bamboo rhizomes was significantly higher during the T1 period compared to the T2 period. These findings indicate that a logging width of 8 m had the highest effect on the accumulation of C elements in various bamboo organs. Moreover, the study highlights that the accumulation of C in different bamboo organs was higher during the peak growth period than that observed at later stages of growth. This phenomenon can be attributed to the rapid growth of various organs and a significant increase in biomass during the peak growth period, resulting in a higher accumulation of C in various organs compared to the late stages of growth. The average accumulation of C in the underground parts (bamboo stumps and rhizomes) was higher than that in the aboveground parts (bamboo leaves and branches), indicating that bamboo allocates more C to the underground parts during the growth period. This allocation strategy indicates that bamboo plants prioritize the growth and expansion of bamboo stumps and rhizomes to access more nutrients and water resources. This finding is consistent with the results reported by Song et al. [43].

During the rapid growth stage, plants require significant nitrogen uptake for cell division, growth, and the synthesis of proteins and carbohydrates [44]. In response to changes in habitat, moso bamboo adjusts its N allocation pattern and N utilization efficiency across the aboveground and underground organs [39]. In this study, bamboo leaves exhibited the highest nitrogen content under the different strip logging modes during the two growth stages compared with the other organs. The nitrogen contents in the leaves, branches, culms, and rhizomes of plants in the logging zones were relatively higher than those in the reserve zones and the Con treatment. This finding indicates that a large amount of nitrogen was allocated for leaf development during the growth stage to enhance photosynthesis intensity and secure adequate light resources for growth [45]. However, the reduction in tree density in the bamboo stands due to selective logging facilitates increased light availability for bamboo leaves, increasing photosynthesis intensity, and promoting N accumulation in most bamboo organs [44]. The average N contents in various organs were higher during the T2 period than during the T1 period, consistent with the results reported by Wang [46]. This could be attributed to a higher accumulation of N in each organ during the late growth stage compared to the peak growth period. During the late growth period, adequate N uptake facilitates growth, metabolic activity, and colonization in various organs.

Phosphorus is an essential component of several macromolecular structures, such as nucleic acid, nuclear proteins, and phospholipids. It is implicated in physiological and metabolic processes such as photosynthesis and respiration, plant cell division and growth, organ development, and plant stress response [40]. In this study, the highest P content was observed in bamboo leaves across different strip logging treatments, consistent with the findings reported by Su et al. [47]. This phenomenon can be attributed to the robust growth of bamboo leaves, where phosphorus participates in various metabolic processes.

## 5. Conclusions

The findings indicate that bamboo leaves, which are the most metabolically active tissue, have the highest phosphorus content. In addition, the P contents in aboveground organs (bamboo leaves, branches, and culms) under different strip logging treatments were significantly higher during the peak growth period compared to the late growth period. However, there were no significant differences in the overall P content in underground organs (bamboo stumps and rhizomes) across the different strip logging treatments.

## Figures and Tables

**Figure 1 plants-13-01448-f001:**
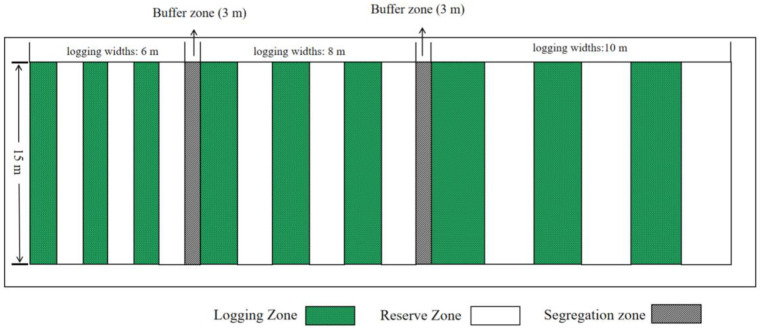
Schematic diagram of establishment of plots.

**Figure 2 plants-13-01448-f002:**
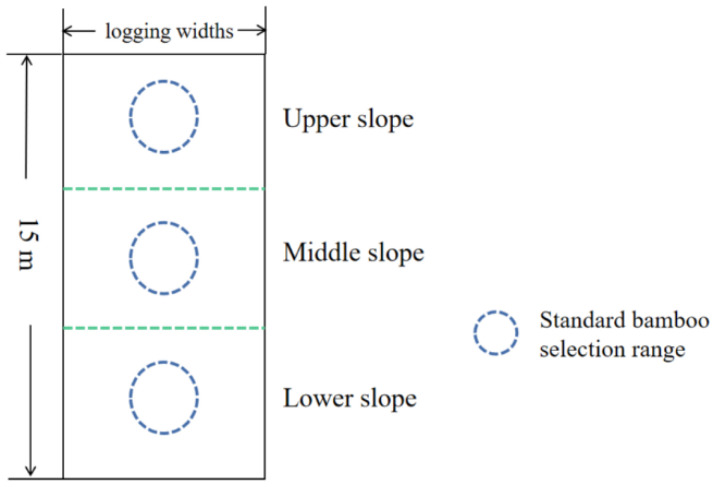
Sampling range of different slope locations within the sample plots.

**Figure 3 plants-13-01448-f003:**
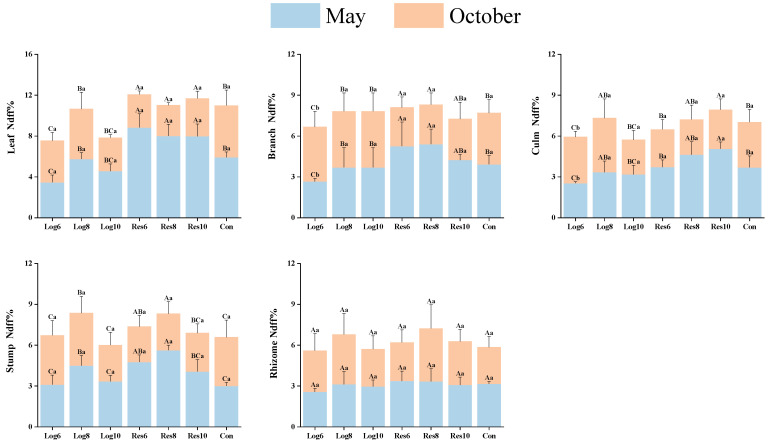
Differences in nitrogen accumulation ability of various organs of *Phyllostachys edulis* under different logging widths of logging zones. Note: The capital letters indicate significant differences in the nitrogen accumulation abilities of the organs of *P. edulis* across different logging widths of logging zones during the same period; lowercase letters indicate significant differences in the nitrogen accumulation abilities of the organs of *P. edulis* between different periods under the same logging width of logging zone. The error line represents the standard error.

**Figure 4 plants-13-01448-f004:**
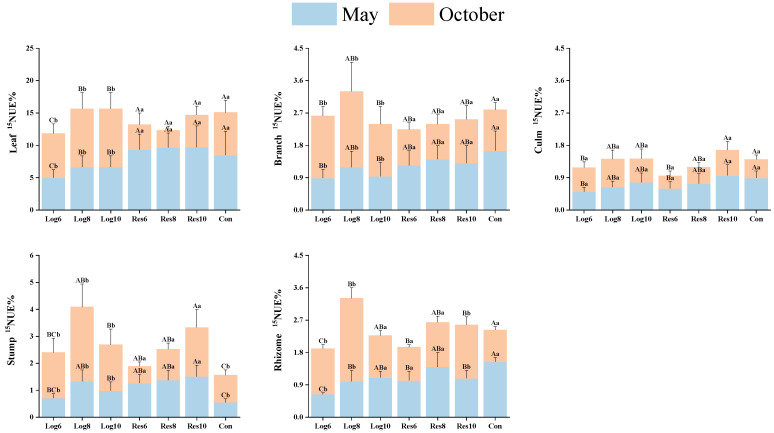
Differences in nitrogen utilization efficiency of different organs of bamboo under different logging widths of logging zones. Note: The capital letters indicate significant differences in the nitrogen accumulation abilities of the organs of *P. edulis* across different logging widths of logging zones during the same period; lowercase letters indicate significant differences in the nitrogen accumulation abilities of the organs of *P. edulis* between different periods under the same logging width of logging zone. The error line represents the standard error.

**Figure 5 plants-13-01448-f005:**
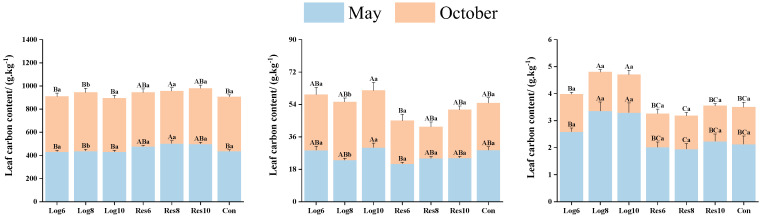
Differences in the contents of key elements in moso bamboo leaves across logging zones with different logging widths. Note: The capital letters indicate significant differences in the nitrogen accumulation abilities of the organs of *P. edulis* across different logging widths of logging zones during the same period; lowercase letters indicate significant differences in the nitrogen accumulation abilities of the organs of *P. edulis* between different periods under the same logging width of logging zone. The error line represents the standard error.

**Figure 6 plants-13-01448-f006:**
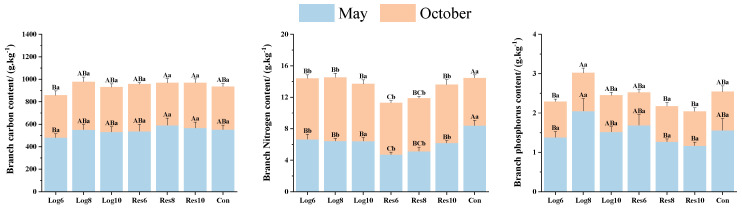
Differences in element composition of bamboo branches in logging zones with different logging widths. Note: The capital letters indicate significant differences in the nitrogen accumulation abilities of the organs of *P. edulis* across different logging widths of logging zones during the same period; lowercase letters indicate significant differences in the nitrogen accumulation abilities of the organs of *P. edulis* between different periods under the same logging width of logging zone. The error line represents the standard error.

**Figure 7 plants-13-01448-f007:**
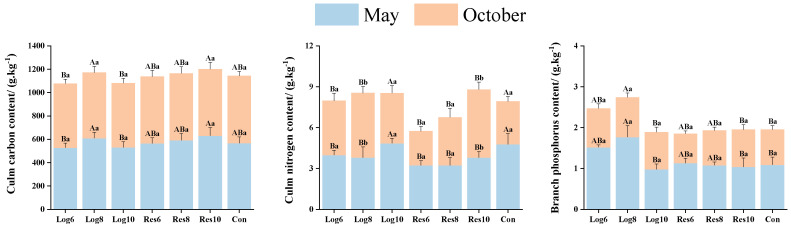
Variations in the elemental composition of bamboo culms across logging zones with different logging widths. Note: The capital letters indicate significant differences in the nitrogen accumulation abilities of the organs of *P. edulis* across different logging widths of logging zones during the same period; lowercase letters indicate significant differences in the nitrogen accumulation abilities of the organs of *P. edulis* between different periods under the same logging width of logging zone. The error line represents the standard error.

**Figure 8 plants-13-01448-f008:**
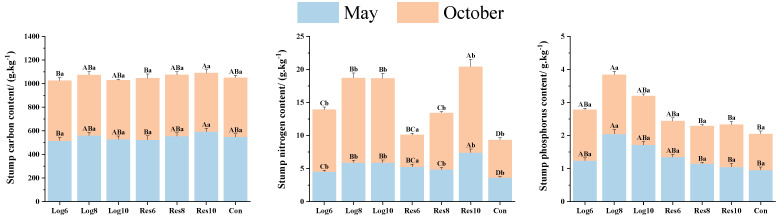
Elemental composition in bamboo stumps in logging zones with varying logging widths. Note: The capital letters indicate significant differences in the nitrogen accumulation abilities of the organs of *P. edulis* across different logging widths of logging zones during the same period; lowercase letters indicate significant differences in the nitrogen accumulation abilities of the organs of *P. edulis* between different periods under the same logging width of logging zone. The error line represents the standard error.

**Figure 9 plants-13-01448-f009:**
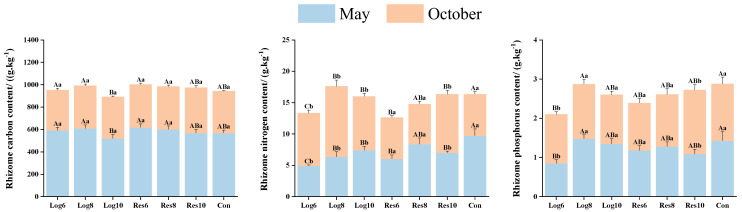
Differences in elemental composition of bamboo rhizomes in logging zones with different logging widths. Note: The capital letters indicate significant differences in the nitrogen accumulation abilities of the organs of *P. edulis* across different logging widths of logging zones during the same period; lowercase letters indicate significant differences in the nitrogen accumulation abilities of the organs of *P. edulis* between different periods under the same logging width of logging zone. The error line represents the standard error.

**Table 1 plants-13-01448-t001:** Stoichiometric characteristics of bamboo leaves in logging zones with different logging widths.

Stoichiometric Characteristics	Period	Log6	Log8	Log10	Res6	Res8	Res10	Con
C/N	May	15.18 ± 1.84 ^Ba^	18.87 ± 2.80 ^ABa^	14.34 ± 0.97 ^Ba^	22.93 ± 3.36 ^Aa^	20.83 ± 0.89 ^Ab^	20.54 ± 1.6 ^Aa^	15.22 ± 1.06 ^Ba^
	October	15.42 ± 1.31 ^Ba^	15.99 ± 0.26 ^Ba^	14.32 ± 0.09 ^Ba^	19.5 ± 1.31 ^ABa^	26.35 ± 4.61 ^Aa^	18.32 ± 3.51 ^Ba^	18.01 ± 0.53 ^Ba^
N/P	May	11.27 ± 2.44 ^ABb^	8.32 ± 3.74 ^Bb^	11.17 ± 6.92 ^ABb^	10.49 ± 1.34 ^ABb^	13.13 ± 3.52 ^Aa^	11.74 ± 3.61 ^ABb^	13.66 ± 1.79 ^Ab^
	October	22.29 ± 0.20 ^Aa^	22.06 ± 0.64 ^Aa^	23.00 ± 0.41 ^Aa^	19.18 ± 0.37 ^ABa^	14.23 ± 2.16 ^Ba^	20.34 ± 3.04 ^Aa^	19.04 ± 0.77 ^ABa^

Note: The capital letters indicate significant differences in C/N and N/P ratios in the organs of *P. edulis* between logging zones with different logging widths during the same period; lowercase letters indicate significant differences in C/N and N/P ratios in the organs of *P. edulis* between different periods under logging zone with the same logging width.

**Table 2 plants-13-01448-t002:** Stoichiometric characteristics of bamboo branches across logging zones with varying logging widths.

Stoichiometric Characteristics	Period	Log6	Log8	Log10	Res6	Res8	Res10	Con
C/N	May	72.31 ± 8.74 ^BCa^	85.57 ± 2.17 ^Ba^	82.8 ± 6.53 ^Ba^	113.71 ± 5.96 ^Aa^	115.68 ± 16.45 ^Aa^	91.69 ± 4.07 ^Aab^	65.88 ± 5.62 ^Ca^
	October	48.46 ± 1.84 ^Bb^	58.81 ± 4.02 ^ABb^	49.36 ± 1.33 ^Bb^	63.78 ± 3.02 ^Ab^	56.01 ± 0.64 ^ABb^	54.12 ± 2.76 ^ABb^	63.03 ± 2.61 ^Aa^
N/P	May	4.92 ± 0.61 ^Bb^	3.23 ± 0.60 ^BCb^	4.25 ± 0.37 ^Bb^	2.81 ± 0.14 ^Cb^	4.10 ± 0.72 ^Bb^	5.36 ± 0.82 ^ABb^	6.61 ± 2.97 ^Aa^
	October	8.51 ± 0.22 ^Aa^	7.49 ± 0.31 ^ABa^	8.70 ± 0.05 ^Aa^	7.92 ± 0.40 ^ABa^	7.49 ± 0.11 ^ABa^	8.51 ± 0.26 ^Aa^	6.20 ± 0.28 ^Ba^

Note: The capital letters indicate significant differences in C/N and N/P ratios in the organs of *P. edulis* between logging zones with different logging widths during the same period; lowercase letters indicate significant differences in C/N and N/P ratios in the organs of *P. edulis* between different periods under logging zone with the same logging width.

**Table 3 plants-13-01448-t003:** Stoichiometric characteristics of bamboo culms in logging zones with different logging widths.

Stoichiometric Characteristics	Period	Log6	Log8	Log10	Res6	Res8	Res10	Con
C/N	May	132.52 ± 15.36 ^Ba^	160.52 ± 2.99 ^ABa^	109.44 ± 6.51 ^Ca^	174.99 ± 15.73 ^Ab^	187.70 ± 41.11 ^Aa^	165.84 ± 11.55 ^ABa^	118.48 ± 3.83 ^BCb^
	October	137.98 ± 7.64 ^BCa^	153.8 ± 14.06 ^Ba^	116.18 ± 0.99 ^Ca^	228.84 ± 16.04 ^Aa^	170.19 ± 54.88 ^Ba^	114.36 ± 3.54 ^Cb^	183.97 ± 14.98 ^Ba^
N/P	May	2.63 ± 0.20 ^Cb^	2.63 ± 1.21 ^Cb^	5.12 ± 1.10 ^Aa^	2.93 ± 0.51 ^Ca^	3.00 ± 0.35 ^Cb^	4.06 ± 1.39 ^Bb^	4.68 ± 1.32 ^ABa^
	October	4.19 ± 0.15 ^Ba^	3.79 ± 0.6 ^BCa^	5.18 ± 0.12 ^Aa^	3.50 ± 0.30 ^Ca^	4.16 ± 0.86 ^Ba^	5.50 ± 0.24 ^Aa^	3.64 ± 0.14 ^BCb^

Note: The capital letters indicate significant differences in C/N and N/P ratios in the organs of *P. edulis* between logging zones with different logging widths during the same period; lowercase letters indicate significant differences in C/N and N/P ratios in the organs of *P. edulis* between different periods under logging zone with the same logging width.

**Table 4 plants-13-01448-t004:** Stoichiometric characteristics of bamboo stumps in logging zones with different logging widths.

Stoichiometric Characteristics	Period	Log6	Log8	Log10	Res6	Res8	Res10	Con
C/N	May	113.44 ± 1.02 ^Ba^	94.85 ± 3.98 ^BCa^	89.69 ± 5.19 ^Ca^	99.49 ± 5.68 ^BCa^	114.35 ± 6.76 ^Ba^	79.73 ± 5.51 ^Ca^	150.45 ± 1.53 ^Aa^
	October	39.73 ± 2.44 ^Cb^	54.72 ± 2.08 ^Bb^	39.13 ± 2.15 ^Cb^	97.91 ± 4.61 ^Aa^	60.61 ± 1.64 ^Bb^	38.31 ± 1.12 ^Cb^	87.97 ± 6.13 ^Ab^
N/P	May	3.68 ± 0.30 ^BCa^	2.90 ± 0.24 ^Cb^	3.44 ± 0.14 ^BCb^	3.94 ± 0.42 ^Ba^	4.29 ± 0.61 ^Bb^	7.16 ± 0.74 ^Ab^	3.82 ± 0.32 ^BCb^
	October	4.29 ± 0.19 ^Ca^	5.24 ± 0.08 ^Ca^	8.65 ± 0.10 ^Ba^	4.43 ± 0.08 ^Ca^	7.53 ± 0.03 ^Ba^	10.15 ± 0.08 ^Aa^	5.22 ± 0.13 ^Ca^

Note: The capital letters indicate significant differences in C/N and N/P ratios in the organs of *P. edulis* between logging zones with different logging widths during the same period; lowercase letters indicate significant differences in C/N and N/P ratios in the organs of *P. edulis* between different periods under logging zone with the same logging width.

**Table 5 plants-13-01448-t005:** Stoichiometric characteristics of bamboo rhizomes in logging zones with different logging widths.

Stoichiometric Characteristics	Period	Log6	Log8	Log10	Res6	Res8	Res10	Con
C/N	May	120.17 ± 6.71 ^Aa^	95.91 ± 3.99 ^Ba^	79.86 ± 5.40 ^BCa^	103.45 ± 16.19 ^Ba^	71.75 ± 0.47 ^Ca^	80.91 ± 5.40 ^BCa^	58.47 ± 3.43 ^Da^
	October	42.79 ± 6.23 ^Bb^	33.93 ± 4.06 ^Cb^	43.32 ± 3.59 ^Bb^	58.30 ± 6.94 ^Ab^	59.75 ± 3.89 ^Ab^	43.44 ± 6.48 ^Bb^	56.70 ± 3.40 ^Aa^
N/P	May	5.80 ± 0.17 ^Ba^	4.29 ± 0.05 ^Cb^	5.52 ± 0.61 ^Bb^	5.09 ± 0.43 ^BCa^	6.54 ± 0.10 ^Aa^	6.48 ± 0.55 ^Aa^	6.80 ± 0.30 ^Aa^
	October	6.71 ± 0.09 ^Ba^	8.06 ± 0.05 ^Aa^	6.87 ± 0.33 ^Ba^	5.46 ± 0.2 ^Ca^	4.80 ± 0.32 ^Db^	5.72 ± 0.07 ^Ca^	4.69 ± 0.99 ^Db^

Note: The capital letters indicate significant differences in C/N and N/P ratios in the organs of *P. edulis* between logging zones with different logging widths during the same period; lowercase letters indicate significant differences in C/N and N/P ratios in the organs of P. edulis between different periods under logging zone with the same logging width.

**Table 6 plants-13-01448-t006:** Principal component analysis of N utilization and nutrient content in various organs of moso bamboo under different strip logging treatments.

Parameters	Principal Components				
	1	2	3	4	5
Leaf Ndff%	0.781	−0.483	−0.135	−0.266	−0.03
Branch Ndff%	0.702	−0.559	0.29	−0.262	0.19
Culm Ndff%	0.919	−0.037	−0.292	−0.005	0.263
Stump Ndff%	0.744	−0.206	0.578	0.145	0.017
Rhizome Ndff%	0.813	−0.236	0.262	0.224	−0.183
Leaf ^15^NUE%	0.622	0.504	−0.174	−0.496	−0.018
Branch ^15^NUE%	0.415	0.711	0.372	−0.297	0.301
Culm ^15^NUE%	0.321	0.685	−0.642	0.072	0.097
Stump ^15^NUE%	0.537	0.633	0.182	0.513	−0.041
Rhizome ^15^NUE%	0.805	0.516	0.142	−0.035	−0.105
Leaf C contents	0.804	−0.322	−0.123	0.417	0.146
Leaf N contents	−0.561	0.813	−0.051	−0.075	0.087
Leaf P contents	−0.155	0.9	0.284	0.104	−0.262
Branch C contents	0.865	−0.079	−0.107	−0.056	−0.473
Branch N contents	−0.132	0.859	−0.084	−0.209	0.433
Branch P contents	0.182	0.464	0.648	−0.494	−0.237
Culm C contents	0.954	−0.119	−0.236	0.006	0.107
Culm N contents	0.036	0.899	−0.365	0.134	0.194
Culm P contents	0.145	0.595	0.663	0.108	0.418
Stump C contents	0.95	−0.029	−0.231	0.184	0.09
Stump N contents	0.251	0.667	−0.213	0.639	−0.148
Stump P contents	0.049	0.727	0.566	0.226	−0.283
Rhizome C contents	0.676	−0.471	0.425	0.077	0.273
Rhizome N contents	0.518	0.761	−0.268	−0.188	−0.123
Rhizome P contents	0.685	0.379	−0.306	−0.452	−0.262
Eigenvalue	9.639	8.169	3.154	2.066	1.311
Contribution rate/%	38.557	32.678	12.618	8.264	5.242
Accumulative contribution/%	38.557	71.234	83.852	92.116	97.358

**Table 7 plants-13-01448-t007:** A comprehensive evaluation of the effects of different strip logging treatments on N utilization and nutrient content in various organs of moso bamboo.

Treatments	Comprehensive Scores	Comprehensive Rank
Log6	−3.93	7
Res6	−3.67	6
Log8	7.24	1
Res8	0.37	3
Log10	−1.78	5
Res10	3.13	2
Con	−1.36	4

## Data Availability

Data are contained within the article.

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
