# Peer review of "Variation in Nitrogen Utilization and Nutrient Composition across Various Organs under Different Strip Logging Management Models in Moso Bamboo (Phyllostachys edulis) Forest"

_plants, 2024, doi:10.3390/plants13111448_

Round 1
Reviewer 1 Report
Comments and Suggestions for Authors
Overall, the authors need to provide a more focused manuscript. The introduction needs to be expanded to provide readers with more background and a clear hypothesis(es)/objective(s) need to be presented. The current “aim of this study to find optimal bandwidth” is too vague. The authors are only looking at a single aspect of N accumulation/utilization, so I would say that they aren’t finding the optimal bandwidth, instead looking at one aspect of logging strip width. They need to present a much more focused view/purpose of their study.
The results section is seemingly endless. The authors provide every statistical difference that exists. I think it is the lack of focus on a hypothesis that creates this effect. I think if the authors were to create a succinct hypothesis/objective that it would allow for paring down of the results section. Just because something is statistically different, doesn’t mean it is meaningful.
Finally, the discussion needs to be expanded. There are so many studies on strip cutting of moso bamboo, but hardly any are references and not in any meaningful way. The authors need to compare/contrast their results with the other studies. If the authors really do want their manuscript to lead to identifying an optimal bandwidth of stirp logging, then they can present a synthesis of other studies with their own results in the discussion to explain the tradeoffs of too skinny or too wide of a strip cuts.
Some specific comments:
Line 28: Authors use the term “bandwidth” in the abstract, but they haven’t defined it. I suggest using a generic term like “logging width treatments” in the abstract unless they want to define bandwidth in the abstract.
Line 35: “C2 and B3 zones had the highest comprehensive evaluation scores” You haven’t defined what this means for the reader. Please reword so that readers can understand based on the text you have presented thus far in the abstract.
Line 36: “In summary, logging zone C2 and reserve zone B3 36 demonstrated greater efficacy in promoting nitrogen utilization and nutrient accumulation across 37 various bamboo organs.” Again I don’t know what treatment logging zone C2 is.
Introduction:
The focus of this manuscript is discussing nitrogen in different organ tissues of the focus species. However, the authors haven’t provided readers enough background information in the introduction about why different levels of nitrogen in the plant organs is important for plant function or in other words, what that difference in nitrogen means at a broader level. I suggest adding another paragraph in introduction that explains what higher or lower nitrogen in the plant organs means for plant function.
Line 83: “combined with indoor analysis” I am unfamiliar with the term indoor analysis. Can the authors provide more context for what indoor analysis is and what it does?
Line 88: Again, the authors use bandwidth, when discussing the logging strip widths. Since bandwidth is term for many types of things, I think the authors can simply say “logging width.”
Line 81-90: It would be great if the authors could provide significant more text to explain their hypothesis, which would help the reader understand how the mechanics of logging strip width might affect soil nitrogen dynamics. For example, do the authors hypothesize that wider logging strips would lead to more erosion and understory vegetation leading to lower accumulated nitrogen in the bamboo? Does having too small of logging width create an environment that negatively affects nitrogen accumulation in the bamboo? The authors should add text that explains the potential dynamics of logging width on nitrogen.
Line 112: authors do not need to say “a portable GPS and compass were used”
Line 114-119: The authors provide information about the site, but I would argue those are results. So in the methods section please just describe what you measured (and how), but then create a new results paragraph at the start of the results section that provides information such as average dbh, soil pH, soil nutrients. I think this paragraph could provide details about the history of the site. When was the site actually logged? You started measuring in 2021, but how long after was this from the logging. What is logging history? In other words, not only when was this logged last, but when was it logged prior? Before the strip logging was it all logged using conventional selective harvest? Soil processes can be slow, so having a since of time since these important disturbance events will help the reader better understand the background of the experiment.
Line 119: I would start a new paragraph that describes the logging treatment in more detail. For example, when did the actual logging take place? You are assessing the site in February, May, October of 2021, are there trees in the logged area? If so, how big are the trees in the logged area compared to reserve, what is the density of the logged area vs reserve?
Line 123: The authors use a naming convention of B1,B2,B3, C1, C2, C3, which seems more abstract than need be. I think this naming system is too arbitrary and something more direct will help readers remember the treatments. I suggest the authors use something like Log6, Log8, Log 10, Res6, Res8, Res10. Instead of CK for control, why not just “Con”. Also instead of T1 and T2 for the two measurement times, why not just use May and Oct.
Line 133-140: Was just one plant per bandwidth injected with labeled N15? If so, can you directly say that to help the readers. Also- did you select a plant in the middle of each bandwidth? Can you provide details on where the plant was to minimize edge effects from the other bandwidths/treatments.
Lines 144-147: How far were the harvested plants from the initial injected plant? Was this consistent across all treatments?
Figure 1: This figure can be improved to help the reader understand your experimental design. First, I suggest showing how the treatments align with the slope of the land. Secondly, showing your N15 injected plant in the bandwidths along with your sampled trees (using a little icon) and include in the legend.
All figures: The authors line up treatments C1, B1, C2, B2…etc. However, having bars of logged plots next to reserve plots seems illogical. Comparing the logged areas of different widths and reserve areas of different widths is more comparable. While all bars can stay on the same graph, why not arrange such that logged widths 6, 8, 10 are next to each other and reserve 6, 8, 10 are next to each other.
Line 688: “During the T period” missing the number.
Results: The authors provide too much text in the results section. If the authors provide a succinct hypothesis and objective in the introduction, that would help with a greater focus of the results. Right now, it seems if the authors did a whole bunch of statistical comparisons and describe everywhere there is a statistical difference. However, a lot of these differences may not actually be valuable to the readers if they are outside of the focus hypothesis/objective.
Discussion: Overall, the discussion needs to be significantly expanded. I don’t believe there was any discussion comparing focused on C3 treatment. Further, there is very little discussion that references other studies comparing strip logging of the species. In your intro you mention other studies looking at 8 and 15m strips, with increased species richness/biomass. How do your results compared to their results. Readers will be curious if your results are similar or different to other studies under similar treatments. I think an expanded section that synthesizes the three strip cuts in comparison to the conventional selective cut is also needed. Adding to the mechanistic understanding of your results is also needed. For example, the N accumulation of C1 was higher in May than October, but higher in October than May for C2, why? Presumably very different things are happening to lead to this result. The strips are only 2 m different, but yet the authors are saying very different processes are occurring with this change. This should be brought out in the discussion.

Author Response
Hello Professor, first of all, I would like to thank you for your valuable comments on my manuscript. In response to your comments, I have made the following revisions: 1. I have revised the professional terminology regarding strip logging, including the code names of the logging widths, reserve zones and logging zones, etc.; 2. I have supplemented the missing parts of the introduction and the discussion section; 3. I have supplemented the missing descriptive parts of the experimental methodology including the infographics of the sample plots; 4. I have simplified and rewritten the part of the result analyses to focus on the key points of the article.

Reviewer 2 Report
Comments and Suggestions for Authors
Add the scientific name of Moso Bamboo on the title.
The abstract is too long, focus on the main results.
Line 35, what is Ndff%?
The aim of work should be rewritten.
The tables and Figure should be reformated and the writings should be bigger in size.
Improve the conclusion part.
For conclusion
Briefly summarize the main findings of the study in a clear and concise way.
Highlight the practical implications of the findings, especially for sustainable agriculture.
For Language
Some sentences can be shortened or rephrased to be more concise.
Use active voice whenever possible.
Proofread carefully for typos and grammatical errors.
Use consistent formatting throughout the sections.
Comments on the Quality of English Language
Moderate editing of English language required
Author Response
Hello Professor, first of all, I would like to thank you for your valuable comments on my manuscript. In response to your comments, I have made the following revisions: 1. I have revised the professional terminology regarding strip logging, including the code names of the logging widths, reserve zones and logging zones, etc.; 2. I have supplemented the missing parts of the introduction and the discussion section; 3. I have supplemented the missing descriptive parts of the experimental methodology including the infographics of the sample plots; 4. I have simplified and rewritten the part of the result analyses to focus on the key points of the article;5、Add the scientific name of Moso Bamboo on the title

Round 2
Reviewer 1 Report
Comments and Suggestions for Authors
I appreciate the work the authors did to revise the manuscript. I feel the revisions are sufficient for publication.
Reviewer 2 Report
Comments and Suggestions for Authors
Accept in the current format.